# Direct observation of excitonic instability in Ta$_2$NiSe$_5$

Kwangrae Kim[1,2,4], Hoon Kim[1,2,4], Jonghwan Kim[2,3], Changil Kwon [1,2], Jun Sung Kim[1,2] & B. J. Kim [1,2✉]

Coulomb attraction between electrons and holes in a narrow-gap semiconductor or a semimetal is predicted to lead to an elusive phase of matter dubbed excitonic insulator. However, direct observation of such electronic instability remains extremely rare. Here, we report the observation of incipient divergence in the static excitonic susceptibility of the candidate material Ta$_2$NiSe$_5$ using Raman spectroscopy. Critical fluctuations of the excitonic order parameter give rise to quasi-elastic scattering of $B_{2g}$ symmetry, whose intensity grows inversely with temperature toward the Weiss temperature of $T_W \approx 241$ K, which is arrested by a structural phase transition driven by an acoustic phonon of the same symmetry at $T_C = 325$ K. Concurrently, a $B_{2g}$ optical phonon becomes heavily damped to the extent that its trace is almost invisible around $T_C$, which manifests a strong electron-phonon coupling that has obscured the identification of the low-temperature phase as an excitonic insulator for more than a decade. Our results unambiguously reveal the electronic origin of the phase transition.

[1] Department of Physics, Pohang University of Science and Technology, Pohang, South Korea. [2] Center for Artificial Low Dimensional Electronic Systems, Institute for Basic Science (IBS), Pohang, South Korea. [3] Department of Materials Science and Engineering, Pohang University of Science and Technology, Pohang, Republic of Korea. [4]These authors contributed equally: Kwangrae Kim, Hoon Kim. ✉email: bjkim6@postech.ac.kr

One of the most spectacular quantum phenomena is the formation of Bose-Einstein condensate (BEC), which allows the quantum mechanical nature of the electron wave function to manifest as macroscopic observable properties. Prominent examples include superfluidity in atomic gases[1,2] and the Josephson effect in superconductors[3,4]. One such possibility is offered by excitons—bound pairs of electrons and holes—whose small mass and charge neutrality give rise to unique features in their condensed phase, such as vanishing Hall resistance[5,6]. Indeed, exciton BEC has been observed in double quantum well systems, in which excitons can be generated optically[7], electrically[8], or magnetically through quantum Hall states[5,6].

However, realization of exciton BEC in thermal equilibrium turns out to be quite challenging and requires the following materials design considerations[9–12]: (i) The density of electron-hole pairs has to be small enough for Coulomb interaction to be effective yet large enough to induce condensation. This requires fine tuning of the band gap; (ii) The valence and the conduction bands must have vanishing interband matrix element to avoid hybridization between them; (iii) The wave functions of the electrons and the holes should be spatially separated from each other to avoid their quick recombination into photons; (iv) Ideally, the bands should have a direct gap, positive (semiconductor) or negative (semimetal), to avoid unit cell enlargement.

$Ta_2NiSe_5$ is one of the few promising candidates, and, to our best knowledge, is the only one that has an instability at $\mathbf{q} = 0$[13,14]. Finite-$\mathbf{q}$ instabilities, such as the one found in $TiSe_2$, result from an indirect gap[15] and necessarily break the translational symmetry[16]. The latter is better known as a charge density wave insulator. Although the formation of a charge density wave in $TiSe_2$ involves condensation of excitons[17], an accompanying structural phase transition renders it difficult to distinguish it from a Peierls transition[15–17]. The contrast between the two phases is analogous to the distinction between a Mott insulator and a Slater insulator: they become symmetry-wise indistinguishable when the unit cell is doubled by an antiferromagnetic order.

For a similar reason, it is difficult to differentiate the insulating phase of $Ta_2NiSe_5$ from a trivial band insulator[18–38]. At its semimetal-to-insulator transition, $Ta_2NiSe_5$ undergoes a simultaneous structural phase transition[39,40] from the high-temperature orthorhombic ($Cmcm$) phase to the low-temperature monoclinic ($C2/c$) phase (Fig. 1a). The group-subgroup relation between the two space groups implies freezing of a $B_{2g}$ phonon mode at the second-order phase transition. Indeed, softening of the $B_{2g}$ acoustic mode, equivalent to the vanishing $C_{55}$ elastic constant and consequent shearing of the Ta-Ni-Ta quasi-one-dimensional chain (Fig. 1a), has been observed by an inelastic x-ray scattering experiment[30]. In the monoclinic phase, the valence band and the conduction band belong to the same irreducible representation (IR) along the $\Gamma$-$Z$ direction, where they overlap with each other, thus allowing a band gap to open by hybridization between them[35,37]. Thus, as pointed out earlier[14–17,20,23,24,26,33,37,38], the distinction between an excitonic insulator and a band insulator in the presence of a lattice distortion may ultimately be only quantitative. However, if the transition is primarily of electronic origin and the structural distortion is only a secondary effect, the latter may, in principle, be engineered away by a clever materials design. Thus, identification of the main driver of the transition is of fundamental importance and the first step toward realizing the elusive excitonic insulator.

## Results and discussion

To address this issue, we monitor simultaneously the lattice and the electronic sectors across the transition using Raman spectroscopy, exploiting its unique access to symmetry-resolved electronic susceptibilities at the zone center. We start by analyzing Raman-active phonon modes. $Ta_2NiSe_5$ has twenty-four Raman-active optical phonon modes, which belong to either $A_g$ or $B_g$ IRs in the monoclinic $C2/c$ structure, and $A_g$, $B_{1g}$, $B_{2g}$, or $B_{3g}$ IRs in the orthorhombic $Cmcm$ structure. In our experimental geometry (Fig. 1b), only the $A_g$ modes ($A_g$ and $B_{2g}$) are allowed in the monoclinic phase (orthorhombic phase) by Raman selection rules. Among the optical phonons, eleven of them are $A_g$ modes in the low-temperature phase ($C2/c$), which branch into three $B_{2g}$ and eight $A_g$ modes in the high-temperature phase ($Cmcm$). Thus, the number of Raman-active phonon modes remains the same across the transition.

All of the eleven Raman-active phonons are clearly resolved at all measured temperatures from 120 to 550 K. Figure 1c shows three representative unpolarized spectra: above, at, and below the transition. The highest-energy mode is found below 300 cm$^{-1}$, consistent with earlier works[22,33,34,36]. The low-temperature spectrum most clearly resolves all the phonon modes. Although they all formally belong to the $A_g$ IR in the monoclinic phase, the ones that belong to the $B_{2g}$ IR in the orthorhombic phase (mode 2, 5, and 6) have distinct patterns in their azimuth angle ($\Psi$: angle between incident light polarization and the $a$-axis) dependence (Fig. 1d, Supplementary Fig. 1 and Supplementary Table 1), consistent with the changes in the lattice structure being minor[23,33]. Upon approaching $T_C$ (Fig. 1c and e), mode 5 seemingly disappears in the unpolarized spectra, but it is well defined in the polarization-resolved spectrum measured at an azimuth angle that fully suppresses the $A_g$ modes in the cross polarization (inset of Fig. 1c). Similarly, mode 2 displays exceptionally strong damping, but is still visible at all temperatures. We note that these anomalies persist far above the $T_C$ even outside of the temperature range in which the $B_{2g}$ acoustic mode responsible for the structural transition shows significant softening[30]. We will come back to discuss the temperature evolution of the phonon modes in more detail later on.

In addition to the strong phonon anomalies, we observe a remarkable upsurge of a quasi-elastic peak (QEP, marked by an inverted triangle in Fig. 1c) near the phase transition. Upon heating, the QEP emerges near 250 K, reaches its maximum intensity at $T_C$, and gradually decreases at high temperatures as its spectral weight is redistributed to higher energies (Fig. 1c and e). Its tail extends to high energies and combines with mode 2, which partly accounts for its asymmetric lineshape. We note that the QEP in our spectra is distinct from 'central' peaks often observed in structural phase transitions[41–44]; the latter typically has an extremely narrow width of the order of 1 cm$^{-1}$ or smaller, and appears only within a few degrees about $T_C$. For example, the Raman spectra of $LiOsO_3$ show very minor change below 200 cm$^{-1}$ across its second-order structural phase transition[45,46]. Thus, with all the lattice degrees of freedom exhausted, the QEP can only come from electronic scattering.

In fact, electronic Raman scattering is widely observed in many strongly correlated systems[47,48], and in some cases serves as a sensitive indicator of an electronic phase transition. In particular, it has been shown in the context of nematic transition in iron-based superconductors that Raman response functions measure bare electronic susceptibilities without being affected by acoustic phonons[49,50]. This allows Raman scattering to selectively probe only the electronic component of the susceptibility even when the electronic order is strongly coupled to a lattice distortion; specifically, in our case it has been pointed out in a recent theory that excitonic fluctuations can be probed in the $B_{2g}$ channel[35].

In Fig. 2a, b, we use the electronic QEP to follow the system's propensity toward the putative excitonic insulating phase. We present the temperature evolution in terms of Raman conductivity[47] $\chi''/\omega$ in the $B_{2g}$ channel. Provided that $\chi_{B_{2g}}$ is

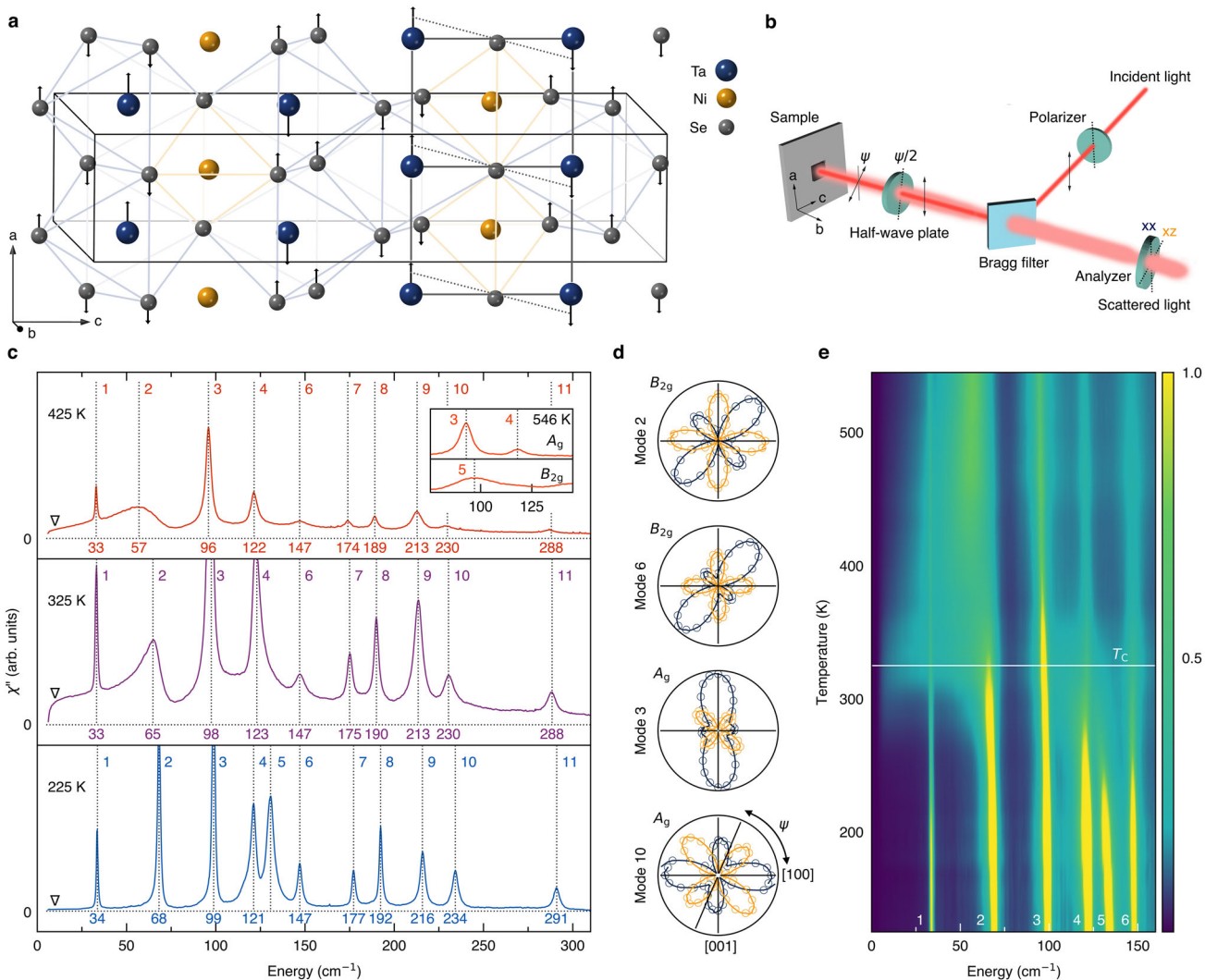

**Fig. 1 Raman-active phonon modes and a quasi-elastic peak. a** Crystal structure of $Ta_2NiSe_5$, overlaid with the displacement vectors (×50) in the orthorhombic-to-monoclinic transition. The black solid box shows the unit cell. The gray solid and dotted lines visualize the shearing of the quasi-one-dimensional Ta-Ni-Ta chain along the $a$-axis by the monoclinic distortion. **b** Experimental configuration of the Raman measurements. **c**, Unpolarized Raman spectra at $T = 225$ K (blue), 325 K (purple), and 425 K (red). All of the eleven Raman-active modes are labeled in the order of increasing energy, and the quasi-elastic peak (QEP) is marked by an inverted triangle. The inset shows symmetry-resolved Raman spectra at $T = 546$ K. **d** Azimuth-angle dependence of the polarization-resolved Raman scattering intensity. Ψ measures the angle between the $a$-axis and incident polarization, and navy (yellow) color represents the scattered-light polarization parallel (perpendicular) to the incident one. The $A_g(B_{2g})$ spectrum in the inset of (**c**) is isolated by measuring in the cross-polarization channel at **Ψ = 45°(Ψ = 0°)**. **e** False color map of the unpolarized Raman spectra from 120 to 550 K. $T_C = 325$ K is indicated by the white horizontal line.

analytic at zero momentum and energy, integration of the Raman conductivity over all energies returns the real part of the uniform static susceptibility[50], which diverges at a thermodynamic phase transition. The QEP contribution to the conductivity increases dramatically as $T_C$ is approached from above (Fig. 2a) and below (Fig. 2b). Figure 2c shows $\chi_{B_{2g}}$ obtained by integrating the conductivity after subtracting phonon contributions. The phonon peaks are fitted with asymmetric Fano lineshape as shown in the inset (see Supplementary Figs. 2 and 3 for the complete fitting result). The resulting static susceptibility is not affected in any significant way by the fitting method because the phonon contributions are small. The result is also in excellent agreement with that obtained using the standard coupled electron–phonon model (Supplementary Fig. 4).

The $\chi_{B_{2g}}$ shows a Curie-Weiss behavior above $T_C$ with the Weiss temperature of $241 \pm 9$ K, extracted from linear extrapolation of the inverse susceptibility (Fig. 2c, d). The Weiss temperature can also be independently estimated from other quantities that are related to the correlation length. Figure 2e shows the inverse amplitude, which is expected to depend linearly on temperature near $T_C$, extracted by fitting to a damped harmonic oscillator lineshape. When extrapolated, they intercept the zero frequency at 241 K, giving a self-consistent quantification of the Weiss temperature. This is the temperature scale of the excitonic insulator transition that would have taken place were it not preempted by the structural phase transition at a higher temperature.

The observation that the Raman susceptibility does not diverge but only is cut off at the $T_C$ implies that it measures the electronic sub-system and not the entire system. This stems from the fact that the electron-acoustic-phonon coupling contribution to the dynamical susceptibility vanishes in the $\mathbf{q} \to 0$ limit, which is a

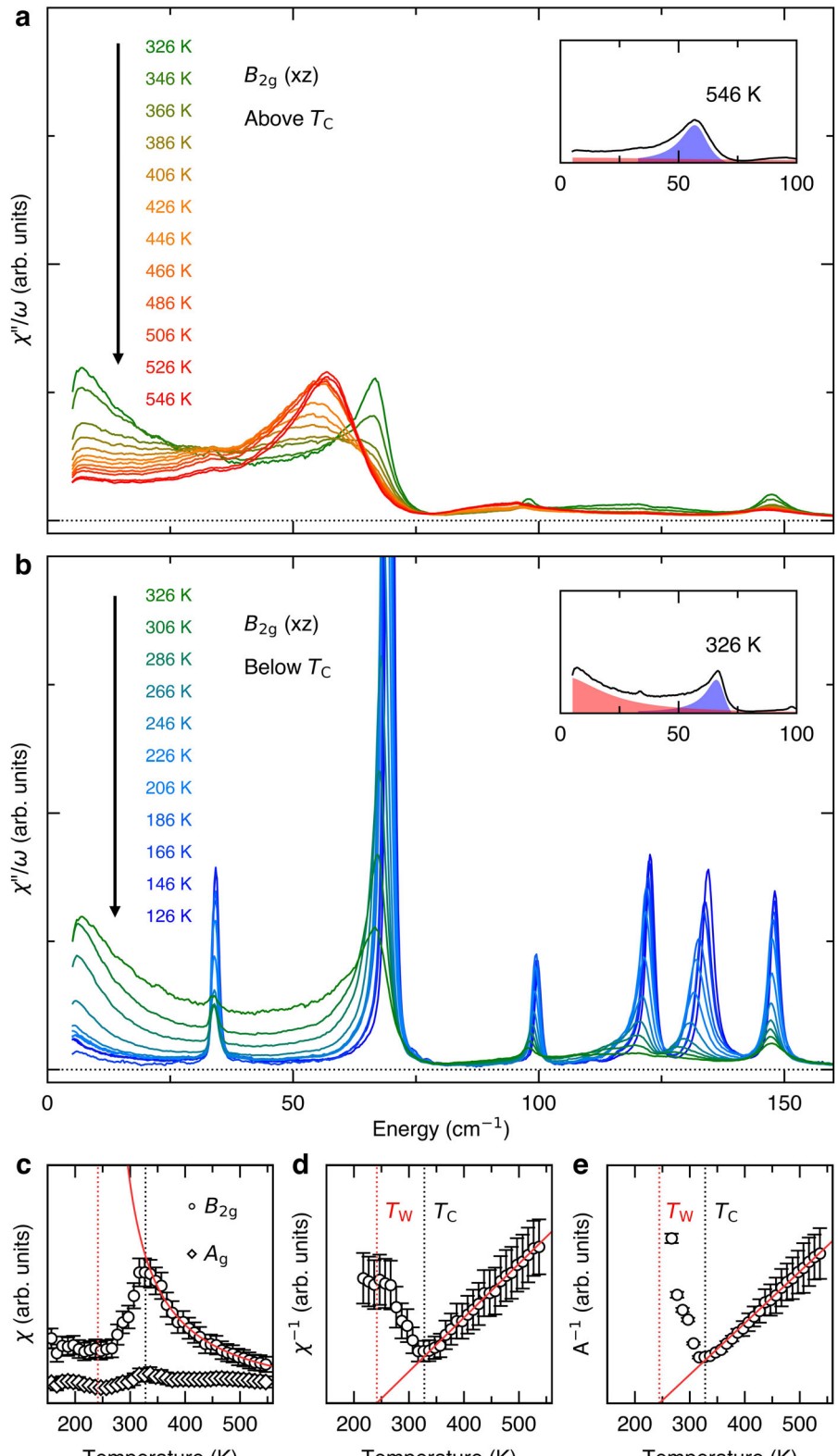

**Fig. 2 Static susceptibility extracted from Raman spectra. a, b** Raman conductivity $\chi''/\omega$ measured above and below the $T_C$, respectively. The insets show representative fitting of the QEP (red) and mode 2 (blue). **c** The real part of the uniform static susceptibility obtained from integrating the Raman conductivity after subtracting the phonon contribution. The $B_{2g}$ susceptibility (circles) follows a Curie-Weiss behavior, whereas the $A_g$ signal (diamonds) is nearly temperature independent. Black and red vertical lines indicate $T_C$ and $T_W$, respectively. **d, e** The temperature dependence of the inverse susceptibility and the inverse amplitude of the QEP, respectively. Red solid lines show the linear fits near the $T_C$. The error bars are defined by one standard deviation for the extracted values.

consequence of the translation symmetry[49,50], or, more precisely, the fact that the symmetry generators associated with acoustic phonons commute with the conserved momentum operator of an electron[51]. This is analogous to Adler's principle for a Lorentz-invariant system[52]; a general criterion for nonrelativistic systems when the coupling between a Nambu-Goldstone boson and Landau quasiparticles vanish in the $\mathbf{q} \rightarrow 0$ limit is given in ref. [51].

Further evidence that the excitonic instability is not driven by the structural instability is provided by the fact that excitonic fluctuations persist up to the highest measured temperature, in contrast to the softening of the $B_{2g}$ acoustic phonon, the lowest frequency mode and hence the main driver of the structural phase transition, that significantly subsides already at 400 K[30].

Electron-lattice couplings, always present in all solid-state systems, have three important consequences in the present case: (i) The $T_C$ is raised in proportion to the square of the coupling strength; (ii) The $C_{55}$ elastic constant for the monoclinic strain (or the corresponding $B_{2g}$ acoustic phonon velocity) is renormalized and vanishes at the $T_C$, which induces a structure distortion[21]. The latter has been observed in a recent inelastic x-ray scattering study[30], but a more detailed temperature dependence of the elastic constant is required for quantitative estimation of the relevant coupling parameters; (iii) The acoustic phase mode of the exciton condensate is gapped out[20].

The critical slowing down of the excitonic fluctuations can also be seen from their interactions and interference with phonons. Figure 3a shows the temperature evolution of the lowest-energy $B_{2g}$ optical phonon (mode 2). As the excitonic fluctuations soften upon approaching $T_C$ from above and sweep through mode 2, its peak amplitude, energy, width, and Fano asymmetry parameter all exhibit strong anomalies (Fig. 3b). In particular, the asymmetry and the width reach a peak above the $T_C$ as its overlap with the continuum excitations maximizes. In contrast, mode 1 remains sharp and almost symmetric, decoupled from the continuum throughout the transition. Remarkably, the asymmetric lineshape of mode 2 remains well below the $T_C$, implying that electronic degrees of freedom within the optical gap is not fully quenched in the insulating phase (Fig. 3c). Interestingly, mode 1 of $A_g$ symmetry acquires small but non-zero asymmetry below $T_C$ as a linear coupling to the continuum[28] becomes allowed in the monoclinic phase (Fig. 3d). The fact that the lineshape is more

symmetric at high temperatures shows that the asymmetry is not simply due to thermal carriers.

Having established the electronic origin of the phase transition, we now discuss competing scenarios of the primary order being predominantly of phononic nature[30,36,37]. A recent DFT study[30,37] proposed that the structural phase transition is driven by an unstable $B_{2g}$ optical phonon mode. In our data, none of the three $B_{2g}$ optical modes shows softening upon cooling in the high-temperature phase. Rather, they become heavily damped to the extent that it is difficult to identify them in the unpolarized spectra. The center-frequency-to-width ratio for mode 2 stays below ~5 above the $T_C$, i.e., it completes only a few oscillation cycles within its lifetime. Let aside the question of softening, it barely fits the notion of a collective excitation.

Mode 5 is even more strongly damped and its weight is broadly distributed in the range 75–150 cm$^{-1}$ close to the $T_C$ (Fig. 4 and Supplementary Fig. 5). This mode has been predicted to be unstable in a recent DFT calculation[36] (see Supplementary Table 2). Below the $T_C$, mode 5 quickly sharpens up and its width becomes comparable to other stable modes by 225 K. In a phonon driven structural phase transition scenario, the strong damping implies a rapid anharmonic decay into a pair of $B_{2g}$ acoustic phonons of opposite momenta[53], which suddenly becomes quenched by the structural phase transition. The renormalization of the $B_{2g}$ acoustic phonon dispersion to zero velocity at $\mathbf{q} = 0$ (ref. 19) reduces the phase space for anharmonic phonon-phonon scattering, and thus the broadening should be minimal near the $T_C$, which is at odds with the experimental observation.

Instead, the exceptionally broad phonon linewidth is naturally accounted for by strong electron–phonon coupling. The phonon rapidly decays into an electron-hole pair, and reciprocally the exciton is dressed by the phonons and acquires a heavy mass. Its spectral weight is largely lost to broad incoherent excitations that extend to high energies. We note that such exciton-phonon complexes have been observed in optical spectra[24,27,29]. As the fluctuations slow down, excitons become trapped to lattice sites and become a static charge transfer from Ni 3d/Se 4p to Ta 5d orbitals, thereby inducing local lattice distortions which eventually trigger a structural phase transition[18,35]. Although initially electronically driven, the eventual electronic and lattice structures are difficult to distinguish from one that results from a band

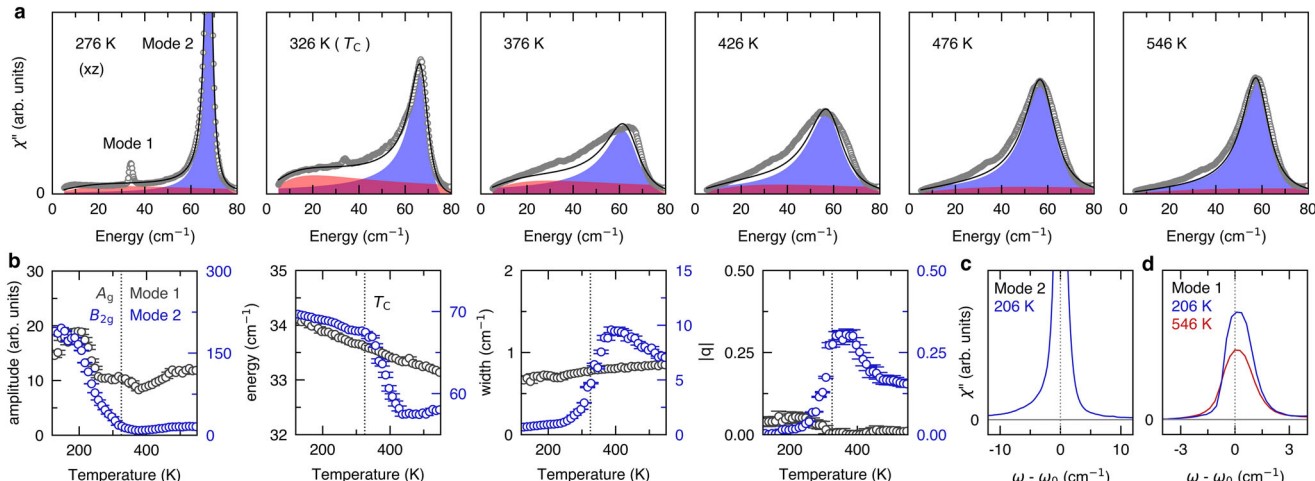

**Fig. 3 Excitonic fluctuations and phonon lineshapes. a** Mode 2 (blue) fitted with the Fano lineshape and QEP (red) with the damped harmonic oscillator lineshape. **b** Peak amplitude, energy, width, and absolute value of Fano asymmetry $q$ extracted from the fitting, and the error bars represent the standard deviation in the fitting procedure. Black dashed line indicates $T_C$. **c** Mode 2 remains asymmetric well below $T_C$. **d** Mode 1 also becomes asymmetric in the low-temperature insulating phase. Black dashed lines in (**c**) and (**d**) represent the center energy of Mode 2 and Mode 1, respectively.

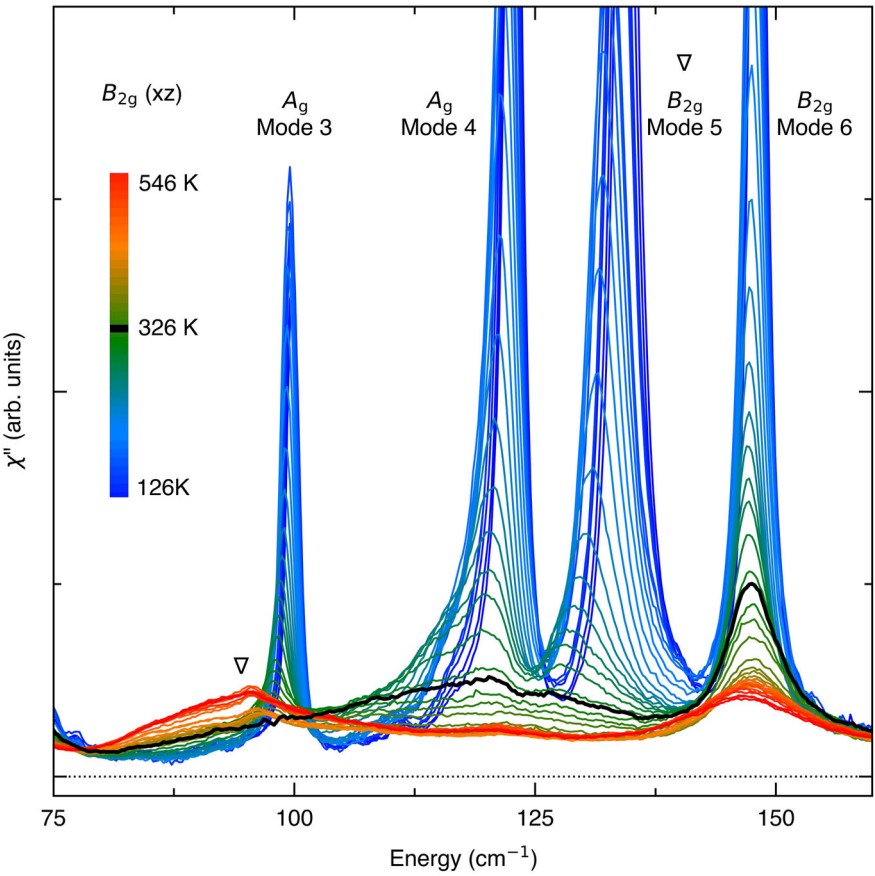

**Fig. 4 Heavily damped $B_{2g}$ phonons.** $B_{2g}$ Raman spectra from 126 K to 546 K. Mode 5 (inverted triangle) exhibits a large blue shift upon cooling, and is heavily damped close to the $T_C$ (black solid line). At the highest measured temperature, all three $B_{2g}$ modes are identified: mode 2 (Fig. 3), 5, and 6. Below $T_C$, $A_g$ modes 3 and 4 also become allowed.

hybridization driven by a structural distortion. Nevertheless, our result unambiguously shows that the phase transition is of electronic origin. Thus, the possibility to realize a pure excitonic insulator is still open, if structural distortions can be suppressed by suitable means, e.g., epitaxial strain.

Upon completion of this work, we became aware of the recent papers in refs. [54–57]. Our work is consistent with the papers in refs. [54] and[57], but not with the papers in ref. [55] and [56]. In particular, the paper in ref. [57] conclude an excitonic origin of the phase transition based on their observation of critical charge fluctuations in their Raman spectra similar to ours, which is missed in ref. [56].

## Methods

**Sample growth**. High-quality single crystals of $Ta_2NiSe_5$ are grown by a chemical vapor transport method. A mixture of Ta, Ni, and Se powder is sintered at 900 °C for 7 days in an evacuated quartz tube. Then, the prepared $Ta_2NiSe_5$ powder is sealed in a quartz tube together with iodine as the transport agent and placed in a furnace with a temperature gradient between 950° and 850°. After 10 days, single crystals (a typical size of 50 μm × 1 mm × 5 mm) are obtained. Their crystallinity and stoichiometry are confirmed by X-ray diffraction and energy-dispersive spectroscopy. The crystals are further characterized by the in-plane resistivity measurements using a Physical Properties Measurement System (Quantum Design). The transition temperature $T_C =$ 325 K was identified by a clear kink in the resistivity, consistent with previous reports[39,40].

**Raman spectroscopy**. The Raman spectra are measured using a home-built Raman spectroscopy setup equipped with a 750-mm monochromator and a liquid-nitrogen-cooled CCD (Princeton Instruments) with a 633-nm He–Ne laser as the excitation source. The elastic light is removed by a set of grating-based notch filters

(Optigrate, BragGrate™ Notch Filter). The home-built setup allows the investigation of low-energy signals (above 5 cm$^{-1}$) with high energy resolution (~1 cm$^{-1}$) in various polarization channels. The samples are mounted in an open-cycle cryostat (Oxford Instruments) with a temperature range from 70 to 500 K. The measurements are conducted in a backscattering geometry where the light propagates along the $b$ crystallographic axis. Raman response of $A_g$ symmetry of the orthorhombic phase are measured in -y(xx)y and -y(x'z')y configurations, and $B_{2g}$ symmetry in -y(x'x')y and -y(xz)y configurations. We use an achromatic half-waveplate to continuously rotate the light polarization between these configurations. The laser power and the beam-spot size are 1.85 mW and ~2 μm, respectively, which give laser heating of 50 K according to the Stokes and anti-Stokes relation of phonons. The corrected temperature and the critical behaviors are consistent with $T_C$, and all the spectra are Bose-corrected. Both spectral variation in grating efficiency and silicon quantum efficiency is not accounted for, which can give an intensity change of <0.1 % in total.

## Data availability

The data supporting the findings of this study are available from the corresponding author upon reasonable request.

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

## Acknowledgements

We thank A. Subedi, G. Y. Cho, E.-G. Moon, K.-Y. Choi, H. W. Yeom, and K.-S. Kim for useful discussions. This project is supported by IBS-R014-A2 and National Research Foundation (NRF) of Korea through the SRC (no. 2018R1A5A6075964).

## Author contributions

B. J. K. conceived the project. C. K. and J. S. K. prepared the samples. K. K. H. K., and J. K. conducted Raman spectroscopy. K. K. and H. K. analysed the data. K. K., H. K., and B. J. K. wrote the manuscript with inputs from all authors. B. J. K. supervised and managed the project.

## Competing interests

The authors declare no competing interests.
