## [Peer Review File · Nature Communications]

REVIEWER COMMENTS

Reviewer #1 (Remarks to the Author):

The experimental results are complete, impressive, and well presented; this can be regarded as a definitive Raman study of a material in which Raman spectroscopy gives highly relevant information in both the phonon analysis and the “electronic Raman” sector. Although some Raman data exists already (Ref 23) this paper includes a number of new and stimulating results, and should definitely be published in some form.

However, I do not see this experiment as the “smoking gun” kind of experiment to prove that TNS is an excitonic insulator - there is no “direct observation” here. Besides, any such claim needs to take all experimental data into account, not just one technique. The strongest hint provided here is the build-up of the QEP in Fig 2c above T_c . It is suggested that this quantity is measuring the “system’s propensity toward the putative excitonic insulating phase” . However, this assignment is speculative, and ignores all details about the temperature-dependent electronic structure of the system, which are surely also relevant, as this quantity ultimately depends on (symmetry-allowed) interband electronic transitions. It is no surprise, for instance, that the QEP signal is suppressed below T_c , as the system becomes gapped and no low-energy interband transitions are possible.

In summary, the data deserve publication in a journal such as npj quantum materials or PRB, but I find the interpretation too simplistic and the conclusions too strong and not fully justified from the data; or at least, other explanations cannot be excluded. Therefore I cannot recommend the paper for Nature Communications.

Reviewer #2 (Remarks to the Author):

This paper shows the results of a detailed examination of the temperature dependence of the phase transition using Raman spectroscopy. They discuss quasi-elastic scattering and certainly observe the anomalies associated with phase transitions. However, the importance of optical phonons in B2g mode for the phase transition has already been reported from various experiments, and the existence of electron-lattice interaction is also mentioned in many papers. Furthermore, two published papers on Nature Communication mention exciton interaction and electron-lattice interaction, and it is already known that the phase transition mechanism at $T_c \sim 325K$ is not understood by the ordinary structural phase transition due to lattice deformation.

On the other hand, it is controversial whether this phase transition at high temperature is due to BEC. Recently, the experimental results of dielectric constant and thermopower measurements suggest that the BEC of this system occurs at low temperatures(doi.org/10.7566/JPSJ.88.113706). Perhaps the author of this paper does not know about the results. The views of the research on the phase transition of TC to 325K are almost in agreement. However, there are few reports on the direct observation of condensed excitons. This paper is also an indirect study suggesting an excitonic insulator transition and is not considered "a direct observation".

I think this paper includes valuable results for the study of excitonic insulators, but I recommend that you submit it to a specialized journal such as condensed matter physics.

Reviewer #3 (Remarks to the Author):

This paper presents exciting and high-quality experimental results on a topic of great current interest,

namely whether the observed phase transition in Ta₂NiSe₅ should be thought of as a purely structural one,

or whether an electronic excitonic effect is the driving force.

This issue has been intensively discussed recently, with contradictory conclusions. It is not an easy call:

as shown in several previous works, the excitonic order parameter couples linearly to some of the structural

modes. Hence even if excitonic in nature the transition will always be accompanied by a structural transition.

This 'chicken and egg' question is thus a quantitative one - assessing whether the electronic effects or the

structural effects dominate. It is nonetheless a very important question, since Ta₂NiSe₅ is one of the most appealing platform among bulk materials for realizing the elusive excitonic insulator phenomenon.

The authors address this using a clever idea. Namely that, because of the specific momentum dependence of

the coupling between the Raman tensor and the structural mode, and because a Raman measurement corresponds

to the 'dynamical' limit in which the momentum q is sent to zero first, a measurement of the Raman susceptibility

picks up the electronic contribution to the excitonic susceptibility.

The experimental results are impressive. The authors are able, in contrast to basically simultaneous work such

as Ref. 49, to follow **all** the phonon modes through the transition. Their resolution and signal to noise also appears

to be excellent. As a result, they are able to clearly observe a 'quasi elastic' peak in the Raman conductivity. The authors

convincingly argue that this peak does not correspond to any of the phonon modes. This allows them to claim with evidence

that (i) the excitonic susceptibility diverges as temperature is lowered, this divergence being cutoff by the intervening

structural transition and (ii) that there is no softening of a phonon mode associated with the transition.

This analysis points to electronic excitonic effects being the driving force of the transition, which is the major result

of this paper and an important one in view of the current debate mentioned above and of the fundamental importance

of the excitonic insulating state of matter.

For these reasons, my assessment is that this paper should eventually be accepted for publication in Nature Communications.

However, I would recommend that the authors consider the following points and make appropriate changes before a final decision

is reached:

(1) Truly, the key idea behind the presented work is an adaptation to Ta₂NiSe₅ (and a different symmetry of the order parameter) of

the approach pioneered in Refs 46,47 in the context of iron-based superconductors and the nematic susceptibility. Although these articles

are quoted, they should be quoted more generously and the connection emphasized more. Also, let me point out that using this approach

based on Raman scattering for Ta₂NiSe₅ was explicitly suggested in the concluding paragraph of Ref.24 (a theoretical paper).

(2) The reason why Raman scattering picks up the electronic excitonic susceptibility should be explained in a more pedagogical way in

the body of the paper. Right now, a discussion is included about analytic vs. non-analytic contributions to the Raman tensor - which probably does

not speak to many readers - but the important point, namely why the momentum dependence is such that the electronic contribution is

picked up by Raman scattering is not clearly explained.

(3) Doesn't the strong damping of mode 2 indicate strong electron-phonon interaction? This should be discussed a bit more.

(4) When using the analogy between χ''/ω and a 'conductivity', and also in connection with the Raman tensor being a non-conserved quantity,

the classic paper by BS Shastry on this topic should in my opinion be quoted: Phys Rev Letters 65, 1068 (1990)

(5) Besides Ref.49 (experimental) and Ref.25 (theory), there is another recent preprint that forcefully argues in favour of a structurally driven transition:

Baldini et al. arXiv:2007.02909. The authors should discuss whether their results are consistent with the experimental observations reported

in this paper, although obviously not with the analysis and main conclusion.

(6) Cosmetic remark: an affiliation is missing for the lead author BJ Kim (!)

Once the paper is revised with a convincing attempt to take these comments into account, I would expect that it will be worthy of publication

in Nature Communications.

Reply to Reviewers

Reviewer #1 (Remarks to the Author):

Comment #1

“The experimental results are complete, impressive, and well presented; this can be regarded as a definitive Raman study of a material in which Raman spectroscopy gives highly relevant information in both the phonon analysis and the “electronic Raman” sector. Although some Raman data exists already (Ref 23) this paper includes a number of new and stimulating results, and should definitely be published in some form.”

Our reply

We thank the Reviewer #1 for his/her high appreciation of our work and providing useful comments.

Comment #2

“However, I do not see this experiment as the “smoking gun” kind of experiment to prove that TNS is an excitonic insulator - there is no “direct observation” here. Besides, any such claim needs to take all experimental data into account, not just one technique.”

Our reply

First, we would like to point out that nowhere in our manuscript did we make the claim that TNS is an excitonic insulator. The Reviewer #1 is mistaken about the central message of our paper. As is clearly stated in the first paragraph, our main claim is that, despite strong electron-phonon coupling, which obscures the identification of TNS as an excitonic insulator, the main driving mechanism behind the phase transition is electronic in origin. We agree with the Reviewer #1 that establishing TNS as an excitonic insulator may be a difficult question to conclude from one experiment, but our claim is only on the existence of an electronic instability, which is evident from our raw data without any interpretation. However, as explained in detail in the introduction part of the paper, this does not imply that TNS is an excitonic insulator because the accompanying structural phase renders the system symmetry-wise indistinguishable from a band insulator, but instead opens the possibility of realizing an ideal excitonic insulator by engineering away the lattice distortion.

Comment #3

“The strongest hint provided here is the build-up of the QEP in Fig 2c above T_c . It is suggested that this quantity is measuring the “system’s propensity toward the putative excitonic insulating phase”. However, this assignment is speculative, and ignores all details about the temperature-dependent electronic structure of the system, which are surely also relevant, as this quantity ultimately depends on (symmetry-allowed) inter-band electronic transitions. It is no surprise, for instance, that the QEP signal is suppressed below T_c , as the system becomes gapped and no low-energy inter-band transitions are possible.”

Our reply

It is a well-established fact that Raman scattering measures the imaginary part of symmetry-resolved, dynamical electronic susceptibilities in the $q \rightarrow 0$ limit, in addition to Raman active optical phonon modes. In this limit, which is taken before the $\omega \rightarrow 0$ limit is taken, the contributions from acoustic phonons vanish, as a result of which only the electronic contribution to the susceptibility is selectively probed. The static susceptibility obtained from Kramers-Kronig transformation of this quantity does not diverge at the transition temperature, consistent with the above argument, but instead clearly shows an anomaly associated with the incipient excitonic instability, which is cut off by the structural phase transition. This unique capability of Raman scattering to selectively probe electronic susceptibility is exploited to show, for example, a nematic instability in iron-based superconductors. Thus, our observation of a divergent susceptibility is by itself direct evidence of an electronic instability, regardless of specific details of the band structure. It is direct in the sense that our existence claim requires no interpretation or a theory model. Our argument relies only on the fundamental fluctuation-dissipation theorem and there is no assumption or speculation made whatsoever. By analogy, establishing a metal-insulator transition from resistivity data does not require considering electronic band structure of the system although resistivity clearly depends on it, because the transition would be evident in the raw data.

To show that the quasi-elastic peak cannot be explained by a simple consideration of electronic band structure, we have performed additional measurements on the sister compound Ta₂NiS₅.

FIG. 1. Comparison of B_{2g} Raman spectra of Ta₂NiSe₅ and Ta₂NiS₅. a, Ta₂NiSe₅ shows an upsurge of QEP near T_c (marked by an inverted triangle). b, Ta₂NiS₅ has no change of spectra in same energy range.

The data show that the Ta₂NiS₅ shows similar phonon peaks but no quasi-elastic peak. As clearly shown in Ref. 13, the electronic structures of the two materials are very similar to each other apart from that Ta₂NiS₅ has a larger gap, and thus considering inter-band electronic transitions of the two materials in the non-interacting picture should lead to similar Raman scattering cross sections. At such level of sophistication of the model, considering the microscopic electronic band structure does not explain the spectral difference between the two materials. To obtain possible electronic instabilities, one has to take into account the effect of interactions in some form of approximation, and such calculation has been performed in Ref. 24 using a Hartree-Fock variational wave function with realistic interactions parameters obtained from constrained RPA. The symmetry of the order parameter is taken to be the one that is consistent with the low-temperature monoclinic phase. Taken together, they conclude that slow electronic fluctuations must be visible in a measurement that selectively address the electronic and lattice degrees of freedom, such as Raman scattering, specifically in the B_{2g} symmetry channel.

Our work is fully consistent with the theory prediction. Because the theory considers the interband transitions starting from the ab initio band structures, we simply refer to their work without reproducing the results in our paper, which in our opinion would be redundant. However, we reiterate that even without relying on the theory calculation our data directly shows existence of slow electronic fluctuations, or an electronic instability. Without the theory, the only logical gap in our paper would be that the electronic instability is that of an excitonic one, but given the fact that TNS has been considered as an excitonic insulator candidate for more than a decade and no other symmetry-broken electronic phase is suggested by any experimental data or theory, it is very reasonable to assume that the electronic instability corresponds to the excitonic one. Moreover, the central unresolved issue of TNS has been on whether the transition is of electronic or structural origin, and not on what type of electronic instability.

Comment #4

"In summary, the data deserve publication in a journal such as npj quantum materials or PRB, but I find the interpretation too simplistic and the conclusions too strong and not fully justified from the data; or at least, other explanations cannot be excluded. Therefore, I cannot recommend the paper for Nature Communications."

Our reply

Again, there is no interpretation as an electronic instability is evident in the raw data, and the Reviewer #1 is mistaken about the main conclusion and perhaps this is why he or she thinks it is too strong. In our view, no other explanation can be given to our data, but we would be very open to consider other scenarios if the Reviewer can be more specific about other possibilities.

Reviewer #2 (Remarks to the Author):

Comment #1

"This paper shows the results of a detailed examination of the temperature dependence of the phase transition using Raman spectroscopy. They discuss quasi-elastic scattering and certainly observe the anomalies associated with phase transitions. However, the importance of optical phonons in B2g mode for the phase transition has already been reported from various experiments, and the existence of electron-lattice interaction is also mentioned in many papers."

Our reply

First, we would like to thank the Reviewer #2 for his/her comments. We agree with the Reviewer #2 that the importance of B2g optical modes for the phase transition and electronic-lattice interaction is widely acknowledged. This is precisely the reason that the issue of whether the phase transition is primarily of excitonic nature or is driven by lattice instability remains unresolved after extensive research over a decade. We provide a definite answer to this question that the phase transition is electronically driven, and this is the central message of our paper.

Comment #2

"Furthermore, two published papers on Nature Communication mention exciton interaction and electron-lattice interaction, and it is already known that the phase transition mechanism at TC~325K is not understood by the ordinary structural phase transition due to lattice deformation. On the other hand, it is controversial whether this phase transition at high temperature is due to BEC. Recently, the experimental results of dielectric constant and thermopower measurements suggest that the BEC of this system occurs at low temperatures(doi.org/10.7566/JPSJ.88.113706). Perhaps the author of this paper does not know about the results. The views of the research on the phase transition of TC to 325K are almost in agreement."

Our reply

We do not agree with the Reviewer #2's view that it is already known that the phase transition is not understood by the ordinary structural phase transition due to lattice deformation. As counterexamples, we point to the recent papers published this year in Ref. 25, 53, 54 which claim that the phase transition is driven by lattice deformation. As the Reviewer #3's points out, the issue of whether the transition is of electronic or structural in origin is a difficult outstanding question with many contradicting results. In our view, this is the foremost question that needs to be addressed before one can discuss whether the transition is of BEC or BCS type.

Comment #3

"However, there are few reports on the direct observation of condensed excitons. This paper is also an indirect study suggesting an excitonic insulator transition and is not considered "a direct observation". I think this paper includes valuable results for the study of excitonic insulators, but I recommend that you submit it to a specialized journal such as condensed matter physics."

Our reply

In the presence of electron-phonon coupling, exciton condensation manifests only as breaking of discrete lattice symmetries, i.e. there is no U(1) symmetry breaking, no off-diagonal long range order, no gapless phase mode, no super-transport properties of any kind from the exciton condensate. This

point is discussed in detail in many papers, for example in Ref. 9 and Ref. 24. Thus, the distinction between an excitonic insulator and a band insulator becomes only a quantitative one, and herein lies the crux of the problem. Our claim is on the direct observation of an electronic instability, and not on direct evidence for an excitonic insulator. Our Raman data show critical electronic fluctuations of B_{2g} symmetry, which lead to a divergence in the static electronic susceptibility. We conclude that the electronic instability is the main driver of the phase transition, and because the symmetry of the order parameter is lower than that of the lattice, it inevitably leads to a structural deformation. Despite the passive role of the lattice, aforementioned features of a pure excitonic insulator are no longer observable. Thus, in the absence of such direct indicator of an excitonic insulator, we believe that the observation of an electronic instability is as direct as it can get in regards to the question whether TNS has propensity toward becoming an excitonic insulator; but again, we do not say that it is an excitonic insulator.

Reviewer #3 (Remarks to the Author):

Comment #1

“This paper presents exciting and high-quality experimental results on a topic of great current interest, namely whether the observed phase transition in Ta₂NiSe₅ should be thought of as a purely structural one, or whether an electronic excitonic effect is the driving force. This issue has been intensively discussed recently, with contradictory conclusions. It is not an easy call: as shown in several previous works, the excitonic order parameter couples linearly to some of the structural modes. Hence even if excitonic in nature the transition will always be accompanied by a structural transition. This ‘chicken and egg’ question is thus a quantitative one - assessing whether the electronic effects or the structural effects dominate. It is nonetheless a very important question, since Ta₂NiSe₅ is one of the most appealing platform among bulk materials for realizing the elusive excitonic insulator phenomenon.

Our reply

We thank the Reviewer #3 for his/her high appreciation of our work. We fully agree with Reviewer #3’s assessment of the current status of the field.

Comment #2

“The authors address this using a clever idea. Namely that, because of the specific momentum dependence of the coupling between the Raman tensor and the structural mode, and because a Raman measurement corresponds to the ‘dynamical’ limit in which the momentum q is sent to zero first, a measurement of the Raman susceptibility picks up the electronic contribution to the excitonic susceptibility.

The experimental results are impressive. The authors are able, in contrast to basically simultaneous work such as Ref. 49, to follow *all* the phonon modes through the transition. Their resolution and signal to noise also appears to be excellent. As a result, they are able to clearly observe a ‘quasi elastic’ peak in the Raman conductivity. The authors convincingly argue that this peak does not correspond to any of the phonon modes. This allows them to claim with evidence that (i) the excitonic susceptibility diverges as temperature is lowered, this divergence being cutoff by the intervening structural transition and (ii) that there is no softening of a phonon mode associated with the transition.

This analysis points to electronic excitonic effects being the driving force of the transition, which is the major result of this paper and an important one in view of the current debate mentioned above and of the fundamental importance of the excitonic insulating state of matter. For these reasons, my assessment is that this paper should eventually be accepted for publication in Nature Communications.”

Our reply

We thank the Reviewer #3 for reading through our paper very carefully and having a very accurate understanding of our paper.

Comment #3

“However, I would recommend that the authors consider the following points and make appropriate changes before a final decision is reached:

(1) Truly, the key idea behind the presented work is an adaptation to Ta₂NiSe₅ (and a different symmetry of the order parameter) of the approach pioneered in Refs 46,47 in the context of iron-based superconductors and the nematic susceptibility. Although these articles are quoted, they should be quoted more generously and the connection emphasized more. Also, let me point out that using this approach based on Raman scattering for Ta₂NiSe₅ was explicitly suggested in the concluding paragraph of Ref.24 (a theoretical paper).”

Our reply

Following the Reviewer #3's comment, we have revised the main manuscript to emphasize more explicitly the connection of our work to Refs 47, 48 and also Ref. 24.
(#1 in summary of changed made)

Comment #4

“(2) The reason why Raman scattering picks up the electronic excitonic susceptibility should be explained in a more pedagogical way in the body of the paper. Right now, a discussion is included about analytic vs. non-analytic contributions to the Raman tensor - which probably does not speak to many readers - but the important point, namely why the momentum dependence is such that the electronic contribution is picked up by Raman scattering is not clearly explained.”

Our reply

We have revised the main manuscript to give an explanation to the reason why Raman scattering picks up the electronic contribution to the susceptibility including an argument from the momentum dependence. We also found an excellent pedagogical paper (Ref. 49) which discusses this point in great detail and we refer to this paper instead of reproducing it in our paper. We agree with the Reviewer #3 that the argument about analytic vs. non-analytic does not speak to many readers and thus have removed it. (#3 and #4 in summary of changed made)

Comment #5

“(3) Doesn't the strong damping of mode 2 indicate strong electron-phonon interaction ? This should be discussed a bit more.”

Our reply

We discuss in detail on the strong damping of mode 2 and mode 5 in the last three paragraphs, considering both scenarios where the strong damping results from (i) anharmonic decay into acoustic phonons and (ii) electron-phonon interactions, and conclude that it is due to the latter. We have added a sentence to make this point a bit more explicit. In our opinion, we have discussed to a sufficient extent, but we would be happy to extend the discussions if the Reviewer #3 can suggest any specific direction to discuss further. (#6 in summary of changed made)

Comment #6

“(4) When using the analogy between χ''/ω and a 'conductivity', and also in connection with the Raman tensor being a non-conserved quantity, the classic paper by BS Shastri on this topic should in my opinion be quoted: Phys Rev Letters 65, 1068 (1990)”

Our reply

Following the Reviewer #3's comment, we have quoted the paper in the relevant places.
(#2 in summary of changed made)

Comment #7

“(5) Besides Ref.49 (experimental) and Ref.25 (theory), there is another recent preprint that forcefully argues in favour of a structurally driven transition: Baldini et al. arXiv:2007.02909. The authors should discuss whether their results are consistent with the experimental observations reported in this paper, although obviously not with the analysis and main conclusion”

Our reply

We have revised the “note added” section to include this paper and also arXiv:2003.10799, which we have missed in our first version.
(#7 and #8 in summary of changed made)

Comment #8

“(6) Cosmetic remark: an affiliation is missing for the lead author BJ Kim (!)”

Our reply

We thank the Reviewer #3 for pointing this out. It is corrected.

Summary of changes made

#1. paragraph 9 in revised version

In response to referee #3 comment #3, we have revised as follows

Original version

“In fact, electronic Raman scattering is widely observed in many strongly correlated systems⁴⁵, and in some cases serve as a sensitive indicator of an electronic phase transition, such as the nematic transition in iron-based superconductors^{46,47}. In our case, a recent theoretical analysis showed that excitonic fluctuations can arise in the B2g channel²⁴.”

Revised version

“In fact, electronic Raman scattering is widely observed in many strongly correlation systems^{45,46}, and in some cases serves as a sensitive indicator of an electronic phase transition. In particular, it has been shown in the context of nematic transition in iron-based superconductors that Raman response functions measure bare electronic susceptibilities without being affected by acoustic phonons^{47,48}. This allows Raman scattering to selectively probe only the electronic component of the susceptibility even when the electronic order is strongly coupled to a lattice distortion: specifically, in our case it has been pointed out in a recent theory that excitonic fluctuations can be proved in the B2g channel²⁴.”

#2. paragraph 10 in revised version

In response to referee #3 comment #6, we have revised as follows

Original version

“In Figs. 2a and 2b, we use the electronic QEP to follow the system’s propensity toward the putative excitonic insulating phase. We present the temperature evolution of the Raman conductivity χ''/ω in the B2g channel. Provided that χ_{B2q} is analytic at zero momentum and energy, integration of the Raman conductivity over all energies returns the real part of the uniform static susceptibility⁴⁷, which diverges at a thermodynamic phase transition.”

Revised version

: “In Figs. 2a and 2b, we use the electronic QEP to follow the system’s propensity toward the putative excitonic insulating phase. We present the temperature evolution in terms of Raman conductivity⁴⁵ χ''/ω in the B2g channel. Provided that χ_{B2q} is analytic at zero momentum and energy, integration of the Raman conductivity over all energies returns the real part of the uniform static susceptibility⁴⁸, which diverges at a thermodynamic phase transition.”

#3. paragraph 12 in original version

In response to referee #3 comment #4, we have deleted following paragraph

Original version

“Our analysis above relies on the assumption of the analyticity of B2g at zero momentum and energy, which, however, is not always satisfied. More precisely, nonanalytic contributions to the uniform static susceptibility may be missed in the dynamical susceptibility probed by Raman scattering; the latter (former) is evaluated at $q \rightarrow 0$, before (after) taking the static limit $\omega \rightarrow 0$.”

Revised version

deleted

#4. paragraph 12 in revised version

In response to referee #3 comment #4, we have revised as follows

Original version

“The fact that the Raman susceptibility does not diverge but only is cut off at the T_c implies that it measures the electronic sub-system and not the entire system. This stems from the fact that the electron-acoustic-phonon coupling contribution to the dynamical susceptibility vanishes in the $q \rightarrow 0$ limit due to the translational symmetry^{46,47}. In general, non-analyticity results from an underlying symmetry of the system and the associated conserved quantity: For example, charge susceptibility is non-analytic because of the particle number conservation and the global U(1) symmetry. In contrast, the excitonic instability is not associated with any conserved quantity and hence is reflected in the Raman susceptibility.”

Revised version

“The observation that the Raman susceptibility does not diverge but only is cut off at the T_c implies that it measures the electronic sub-system and not the entire system. This stems from the fact that the electron-acoustic-phonon coupling contribution to the dynamical susceptibility vanishes in the $q \rightarrow 0$ limit, which is a consequence of the translation symmetry^{47,48}, or, more precisely, the fact that the symmetry generators associated with acoustic phonons commute with the conserved momentum operator of an electron⁴⁹. This is analogous to the Adler’s principle for a Lorentz-invariant system⁵⁰; a general criterion for nonrelativistic systems when the coupling between a Nambu-Goldstone boson and Landau quasiparticles vanish in the $q \rightarrow 0$ limit is given in Ref. 49.”

#5. paragraph 14 in revised version

We have revised as follows

Original version

paragraph 11 “Electron-lattice couplings, ~ the exciton condensate is gapped out⁹.”
before paragraph 12 “The fact that the Raman susceptibility does not diverge ~”

Revised version

paragraph 14 “Electron-lattice couplings, ~ the exciton condensate is gapped out⁹.”
after paragraph 12 “The observation that the Raman susceptibility does not diverge ~”

#6. paragraph 17 in revised version

In response to referee #3 comment #5, we have revised as follows

Original version

“In a phonon driven structural phase transition scenario, this implies a rapid an-harmonic decay into a pair of acoustic phonons of opposite momenta⁴⁸, which suddenly becomes quenched by the structural phase transition. In contrast, the renormalization of the acoustic phonon dispersion to zero velocity at $q=0$ (ref. 19) reduces the phase space for an-harmonic phonon-phonon scattering, and thus the broadening should be minimal near the T_c .”

Revised version

“In a phonon driven structural phase transition scenario, the strong damping implies a rapid an-harmonic decay into a pair of acoustic phonons of opposite momenta⁵¹, which suddenly becomes quenched by the structural phase transition. The renormalization of the acoustic phonon dispersion to zero velocity at $q=0$ (ref. 19) reduces the phase space for an-harmonic phonon-phonon scattering, and thus the broadening should be minimal near the T_c , which is at odds with the experimental observation.”

#7. paragraph 18 in revised version

In response to referee #3 comment #7, we have added note as follows

Revised version

“*Note added.* Upon completion of this work, we became aware of the recent papers in Refs. 52-55. Our work is consistent with the papers in Ref. 52 and Ref. 55, but not with the papers in Ref. 53 and Ref. 54. In particular, the paper in Ref. 55 conclude an excitonic origin of the phase transition based on their observation of critical charge fluctuations in their Raman spectra similar to ours, which is missed in Ref. 54.”

#8. references in revised version

In response to referee #3 comment #3, #4, #6, and #7, we have added relevant references as follows

- [45] B. S. Shastry and B. I. Shraiman, Phys. Rev. Lett. **65**, 1068 (1990).
- [49] H. Watanabe and A. Vishwanath, Proceedings of the National Academy of Sciences **111**, 16314 (2014).
- [50] S. L. Adler, Phys. Rev. **137**, B1022 (1965).
- [52] P. Andrich, H. M. Bretscher, Y. Murakami, D. Golez, B. Remez, P. Telang, A. Singh, L. Harnagea, N. R. Cooper, A. J. Millis, P. Werner, A. K. Sood, and A. Rao, (2020), arXiv:2003.10799 [cond-mat.str-el].
- [53] E. Baldini, A. Zong, D. Choi, C. Lee, M. H. Michael, L. Windgatter, I. I. Mazin, S. Latini, D. Azoury, B. Lv, A. Kogar, Y. Wang, Y. Lu, T. Takayama, H. Takagi, A. J. Millis, A. Rubio, E. Demler, and N. Gedik, (2020), arXiv: 2007.02909 [cond-mat.str-el].

REVIEWER COMMENTS

Reviewer #1 (Remarks to the Author):

In my first report I was positive about the quality of the experimental results but did not see this experiment as clearly advancing the field. However, taking into account the authors response to my comments, and also taking into account the detailed report and response to Review #3, would like to change my recommendation that the paper can now be published in Nature Communications.

I will admit that I misrepresented the subtlety between the repeated claim of an “excitonic instability” (e.g. in the title) and the fact that there is no claim that TNS actually is “an excitonic insulator” in the paper. I therefore accept the response to comment #2, and in fact I prefer the interpretation in this paper over the stronger claims in other papers such as Ref 55.

On the comment #3, let me say this as a general point that also applies to the work on the electronic contribution to Raman studied in the context of “nematic” phase of Fe-based superconductors. What I don’t find very convincing, personally, is that the upturn in this quantity, upon approaching the structural transition is a direct demonstration of an electronic instability, as opposed to being a general signature of an upcoming phase transition in that symmetry channel. My point is that the authors do not cite here, and I am not aware of, any counter-examples, i.e. a system which is metallic or semimetallic, which has a second-order $q=0$ structural phase transition that does NOT show this quantity having an enhancement above T_c , and is therefore NOT electronically driven. Without this kind of context, a sceptical reader from outside the Raman scattering community will have a hard time to ever accept the current results as being particularly significant.

However, my general misgivings are probably not widely shared, and the results are likely to be of interest to the wider community. Moreover the proliferation of similar papers and continued work on this topic (see new references) will ensure that this work is widely cited and will have some impact.

It is not especially important for my overall judgement on the paper, but I would like to come back on the point raised by the authors about the S analogue:

“As clearly shown in Ref. 13, the electronic structures of the two materials are very similar to each other apart from that Ta₂NiS₅ has a larger gap, and thus considering inter-band electronic transitions of the two materials in the non-interacting picture should lead to similar Raman scattering cross sections. At such level of sophistication of the model, considering the microscopic electronic band structure does not explain the spectral difference between the two materials.”

While the two systems may be similar according to DFT calculations in Ref 13, their experimental electronic structures are completely distinct, with the S having a band gap (J. Mater. Chem. C, 2018, 6, 3976) while the Se has a narrow band overlap at high temperatures above T_c (Phys. Rev. Research 2, 013236 2020). Resistivity measurements would also tend to support that the S is a semiconductor at all temperatures. Therefore I would contend that “the microscopic electronic band structure

DOES explain the spectral difference between the two materials” – as it is completely trivial that there are no low-energy electronic excitations in any symmetry channel in the semiconducting Ta₂NiS₅. Note, I am not saying the temperature-dependence above T_c can be explained by band structure considerations, but the existence of a signal at all above T_c (compared to Ta₂NiS₅), and its extinction below T_c, can surely be linked to (although not fully explained by) the temperature-dependent gapping of the electronic states.

Reviewer #3 (Remarks to the Author):

I am satisfied with the reply to my previous comments, and also feel that the revised version does improve this paper especially

by placing it in a broader context and quoting the literature more appropriately.

There is however one important point which, in my opinion, requires full clarification before the paper can be fully recommended for publication

in Nature Communications. Namely: the theoretical arguments. developed in refs.47,48,49 apply to an acoustic phonon. If I am not mistaken, the B_{2g} mode 2

on which the present analysis Raman response is based is an *optical* phonon. The authors should detail the theoretical reasons why the same arguments (about the

q-dependence of the electron-lattice coupling in the Raman susceptibility) also apply to this optical phonon. This is a crucial point.

If clarified convincingly, I would expect the paper to deserve publication in Nature Communications.

In several places, the authors discuss the case of an acoustic phonon. This may be misleading if the authors do not explicitly indicate which acoustic mode they are talking about.

Reviewer #4 (Remarks to the Author):

I read carefully the ms of Kim et al., the reports by the 3 reviewers as well as the reply by the authors. Overall, I share the opinion of reviewer 3. This is an outstanding work that harnesses the unique power of Raman scattering to address $q=0$ electronic instabilities in a symmetry resolved way. The paper gives in my opinion a definitive answer to the "electron versus lattice origin" controversy of the transition observed in Ta₂NiSe₅.

In my opinion both reviewers 1 and 2 failed to understand both the rationale and the importance of the work by Kim et al., likely because they read it only superficially. As a consequence their comments remain rather generic and mostly irrelevant.

My recommendation is therefore that this work deserves to be published in Nature Comm.

Nevertheless, I feel few improvements mostly about the analysis of phonon behavior, and clarifications should be done before publication.

(1) When discussing phonon selection rules, polar plots are very visual but are somewhat hard to decipher for a non-expert. In particular to appreciate the A_{1g} versus B_{2g} distinction above and below T_c, theoretical polar plots / fits using the tensors given in the SI would be valuable (in the SI the Raman tensor are given for the high T phase, they should be given for the low T phase also). This is particularly true for mode 5 whose B_{2g} origin appears to be linked to its polar dependence in the low T phase that appears to be reminiscent of its B_{2g} character. More details about the reasoning are needed for the reader here.

(2) Along the same line the photon polarizations used should be indicated on figure 2, 3 and 4.

(3) More details should be given on the fits of figure 3. As I understand the fits were done using a QEP and a Fano lineshape. In principle these two contributions cannot be considered independent and the full coupled electron-phonon coupled response should be modeled. This was done in ref. 55 based on the standard formalism given by M.V. Klein in for e.g. Light Scattering in Solids I. I would recommend the authors to attempt such a fitting procedure in order to clearly rule out a significant contribution of the optical B_{2g} phonon to the extracted susceptibility. I also recommend the authors to show fits at all available T close to T_c (between 300 and 400K) to assess the fit quality (this can be done in the SI). Also the behavior of the QEP width should also be plotted as it should track the inverse of the QEP amplitude close to T_c.

(4) The fact that some phonon anomalies are seen significantly above T_c is intriguing.

Mode 2 in particular starts to harden already at 400K. This might be due to its coupling to the QEP as stated in the text but I am not convinced that the coupling to the QEP alone can account for this effect. This goes back to the previous point: only a full coupled response (QEP + phonon) analysis can demonstrate this.

Whether or not the anomalies of mode 5 also occur significantly above T_c (or at T_c as implied in the text) is also not clear from figure 4. Again color coding is appealing for the eye, but not so much when one wants to analyze in details a temperature dependence. A figure with vertically stacked spectra would help here.

Reply to Reviewers

Reviewer #1 (Remarks to the Author):

Comment #1

"In my first report I was positive about the quality of the experimental results but did not see this experiment as clearly advancing the field. However, taking into account the authors response to my comments, and also taking into account the detailed report and response to Review #3, would like to change my recommendation that the paper can now be published in Nature Communications. I will admit that I misrepresented the subtlety between the repeated claim of an "excitonic instability" (e.g. in the title) and the fact that there is no claim that TNS actually is "an excitonic insulator" in the paper. I therefore accept the response to comment #2, and in fact I prefer the interpretation in this paper over the stronger claims in other papers such as Ref 55."

Our reply

We thank Reviewer #1 for his/her careful reassessment of our work and recommendation of our paper for publication in Nature Communications.

Comment #2

"On the comment #3, let me say this as a general point that also applies to the work on the electronic contribution to Raman studied in the context of "nematic" phase of Fe-based superconductors. What I don't find very convincing, personally, is that the upturn in this quantity, upon approaching the structural transition is a direct demonstration of an electronic instability, as opposed to being a general signature of an upcoming phase transition in that symmetry channel. My point is that the authors do not cite here, and I am not aware of, any counter-examples, i.e. a system which is metallic or semimetallic, which has a second-order $q=0$ structural phase transition that does NOT show this quantity having an enhancement above T_c , and is therefore NOT electronically driven. Without this kind of context, a sceptical reader from outside the Raman scattering community will have a hard time to ever accept the current results as being particularly significant. However, my general misgivings are probably not widely shared, and the results are likely to be of interest to the wider community. Moreover the proliferation of similar papers and continued work on this topic (see new references) will ensure that this work is widely cited and will have some impact."

Our reply

We think this is a very interesting and important point. We agree with Reviewer #1 that for general readership it would be more convincing to provide a counter-example of a metallic or semimetallic system with a second-order $q=0$ structural phase transition NOT showing a QEP enhancement near T_c . We have done an extensive search in the literature and were able to find one well-studied case of LiOsO_3 . This system is metallic above and below its $q=0$ second-order transition at $T=140$ K, which is studied in depth by neutron diffraction (Y. Shi et al., Nature Materials 12, 1024-1027 (2013)). An independent Raman study (F. Jin et al., PNAS 114, 20327 (2019)) shows no sign of anomaly across the transition apart from new phonon peaks that appear in the low-symmetry phase. This paper shows Raman spectra only above 180 cm^{-1} , but the authors were kind enough to provide to us the raw data over the entire measured energy through a private communication. It is clear from the data shown below that the anomaly that we see in our data is not a general signature appearing at all structure transitions. In the revision, we have cited this paper as a counter-example.

Comment #3

“It is not especially important for my overall judgement on the paper, but I would like to come back on the point raised by the authors about the S analogue: “As clearly shown in Ref. 13, the electronic structures of the two materials are very similar to each other apart from that Ta₂NiS₅ has a larger gap, and thus considering inter-band electronic transitions of the two materials in the non-interacting picture should lead to similar Raman scattering cross sections. At such level of sophistication of the model, considering the microscopic electronic band structure does not explain the spectral difference between the two materials.” While the two systems may be similar according to DFT calculations in Ref 13, their experimental electronic structures are completely distinct, with the S having a band gap (J. Mater. Chem. C, 2018, 6, 3976) while the Se has a narrow band overlap at high temperatures above T_c (Phys. Rev. Research 2, 013236 2020). Resistivity measurements would also tend to support that the S is a semiconductor at all temperatures. Therefore I would contend that “the microscopic electronic band structure DOES explain the spectral difference between the two materials” – as it is completely trivial that there are no low-energy electronic excitations in any symmetry channel in the semiconducting Ta₂NiS₅. Note, I am not saying the temperature-dependence above T_c can be explained by band structure considerations, but the existence of a signal at all above T_c (compared to Ta₂NiS₅), and its extinction below T_c, can surely be linked to (although not fully explained by) the temperature-dependent gapping of the electronic states.”

Our reply

We fully agree with Reviewer #1 that the difference in the electronic structure between the two materials is closely related to their spectral difference. After all, our claim is that the anomaly is due to an electronic instability. Our earlier point, however, was that a simple consideration of transition matrix elements would not be sufficient to fully explain the strong temperature dependence. Thus, it is necessary to take into account in some way the effects of electron interactions that underlie the transition. This is beyond the scope of our study but is already presented in Ref. 24. We think that this paper provides a theoretical basis for our observation.

Reviewer #3 (Remarks to the Author):

Comment #1

"I am satisfied with the reply to my previous comments, and also feel that the revised version does improve this paper especially by placing it in a broader context and quoting the literature more appropriately."

Our reply

We are pleased to hear that Reviewer #3 finds that our paper is improved by more appropriately addressing the broad readership.

Comment #2

"There is however one important point which, in my opinion, requires full clarification before the paper can be fully recommended for publication in Nature Communications. Namely: the theoretical arguments developed in refs.47,48,49 apply to an acoustic phonon. If I am not mistaken, the B2g mode 2 on which the present analysis Raman response is based is an *optical* phonon. The authors should detail the theoretical reasons why the same arguments (about the q-dependence of the electron-lattice coupling in the Raman susceptibility) also apply to this optical phonon. This is a crucial point. If clarified convincingly, I would expect the paper to deserve publication in Nature Communications."

Our reply

Reviewer #3 is correct that the theoretical arguments apply to acoustic phonons. This is precisely what our main conclusion is based on. In paragraph 9, we have clearly stated that the electronic component of the susceptibility is probed without being affected by acoustic phonons, and this is important because the transition also involves freezing of a B2g acoustic phonon. This, however, is separate from the B2g optical mode 2 which we analyze later in the paper. Although B2g optical phonons are linearly coupled to the B2g acoustic phonon and also to the B2g electronic mode and therefore their contributions to the electronic susceptibility cannot be ruled out, it is clear that the divergence in the electronic susceptibility cannot be accounted for by optical phonons that remain at finite frequencies all temperatures. Therefore, our main claim is not affected in any significant way by the B2g optical phonons.

That being said, the reason we provide the analysis of the B2g optical phonon mode 2 is that dynamic fluctuations of the unstable electronic mode above T_c do interfere with mode 2 and thus become visible through the resulting Fano lineshape of the mode 2. Our analysis shows that the width and the asymmetry of mode 2 maximize above T_c . We also discuss on the mode 5 at a higher frequency because of the competing scenario based on DFT calculations that predict softening of an optical phonon mode. But again, our main conclusion does not rely on these latter two points.

Comment #3

"In several places, the authors discuss the case of an acoustic phonon. This may be misleading if the authors do not explicitly indicate which acoustic mode they are talking about."

Our reply

In our paper, in all places discussing acoustic phonon but one in paragraph 12 (where we discuss the theoretical aspect in a general context), we refer specifically to the B2g acoustic mode that freezes at the transition. In the introduction part of the paper (paragraph 5), this mode is specified as being equivalent to the vanishing C_{55} elastic constant and responsible for the shearing of Ta-Ni-Ta chain. To avoid any confusion, we have added "B2g" in front of "acoustic phonons" in all places referring to this mode.

Reviewer #4 (Remarks to the Author):

Comment #1

"I read carefully the ms of Kim et al., the reports by the 3 reviewers as well as the reply by the authors. Overall, I share the opinion of reviewer 3. This is an outstanding work that harness the unique power of Raman scattering to address $q=0$ electronic instabilities in a symmetry resolved way. The paper gives in my opinion a definitive answer to the "electron versus lattice origin " controversy of the transition observed in Ta₂NiSe₅. In my opinion both reviewers 1 and 2 failed to understand both the rationale and the importance of the work by Kim et al., likely because they read it only superficially. As a consequence their comments remain rather generic and mostly irrelevant. My recommendation is therefore that this work deserves to be published in Nature Comm."

Our reply

We thank Reviewer #4 for his/her high appreciation of our work and recommendation for publication in Nature Communications.

Comment #2

"(1) When discussing phonon selection rules, polar plots are very visual but are somewhat hard to decipher for a non-expert. In particular to appreciate the A_{1g} versus B_{2g} distinction above and below T_c, theoretical polar plots / fits using the tensors given in the SI would be valuable (in the SI the Raman tensor are given for the high T phase, they should be given for the low T phase also). This is particularly true for mode 5 whose B_{2g} origin appears to be linked to its polar. dependence in the low T phase that appears to be reminiscent of its B_{2g} character. More details about the reasoning are needed for the reader here."

Our reply

Following Reviewer #4's suggestion, we provide in the SI measured azimuth profile for all phonon modes and fits using theoretical Raman tensors above and below T_c. For most of the modes, A_{1g} and B_{2g} characters are evident from their azimuth profiles even below T_c as mixing between different modes is rather small. These azimuth profiles confirm that A_g and B_{2g} modes have zero intensities at $\Psi=0$ and 45 degrees, respectively. This allows clean separation of modes of different symmetries: As shown in the inset of Fig. 1c, mode 5 is identified to be of B_{2g} symmetry because the cross-polarization channel at $\Psi=0$ degrees measures a pure B_{2g} spectrum. Thus, although mode 5 azimuth profile is not clearly of B_{2g} type due to its overlap with mode 3 and 4, there is no ambiguity in assigning it as B_{2g}. This point is explained in the main text and in more detail in the SI.

Comment #3

"(2) Along the same line the photon polarizations used should be indicated on figure 2, 3 and 4."

Our reply

We have added the photon polarizations in figure 2, 3, and 4 following the Reviewer #4's comment.

Comment #4

(3) More details should be given on the fits of figure 3. As I understand the fits were done using a QEP and a Fano lineshape. In principle these two contributions cannot be considered independent and the full coupled electron-phonon coupled response should be modeled. This was done in ref. 55 based on the standard formalism given by M.V. Klein in for e.g. Light Scattering in Solids I. I would recommend the authors to attempt such a fitting procedure in order to clearly rule out a significant contribution of the optical B_{2g} phonon to the extracted susceptibility. I also recommend the authors to show fits at all available T close to T_c (between 300 and 400K) to assess the fit quality (this can be done in the SI). Also the behavior of the QEP width should also be plotted as it should track the inverse of the QEP amplitude close to T_c."

Our reply

We agree with Reviewer #4 that the full coupled electron-phonon mode is required in principle as the two contributions cannot be considered independent. In the SI we now provide comparison of the fitting results obtained by using two different models, and find excellent agreement between the two. The fit quality of the coupled electron-phonon model tends to be not as good as that of the independent electron-phonon model because asymmetric lineshapes are not reproduced in the

former. However, for the purpose of subtracting off phonon contribution for extracting electronic susceptibility, the fitting method does not affect the result in any significant way because the phonon contribution is very small as we have noted earlier. Following Reviewer #4 suggestion, we now show in the SI the fits for all measured spectra, and also show the QEP width which consistently tracks the inverse amplitude.

Comment #5

“(4) The fact that some phonon anomalies are seen significantly above T_c is intriguing. Mode 2 in particular starts to harden already at 400K. This might be due to its coupling to the QEP as stated in the text but I am not convinced that the coupling to the QEP alone can account for this effect. This goes back to the previous point: only a full coupled response (QEP + phonon) analysis can demonstrate this. Whether or not the anomalies of mode 5 also occur significantly above T_c (or at T_c as implied in the text) is also not clear from figure 4. Again color coding is appealing for the eye, but not so much when one wants to analyze in details a temperature dependence. A figure with vertically stacked spectra would help here.”

Our reply

In the new fitting result with the coupled electron-phonon model shown in Fig. S4, it is seen that the hardening of mode 2 starting around 400 K correlates with the rise in the coupling constant at around same temperature, which is consistent with the hardening being due to the coupling to QEP. Mode 5, although not analyzed in detail, shows a similar behavior with even stronger damping, which now can be seen from the vertically stacked spectra in Fig. S5. As we have mentioned earlier, such heavy damping is difficult to explain within a purely structure-driven phase transition scenario, and given that our Raman data shows critical electronic fluctuations, it is natural to attribute it to the strong electron-lattice coupling. We hope that our additional analysis result strengthens the argument.

Summary of changes made

#1. paragraph 7 in revised version

In response to reviewer #4 comment #2, we have revised as follows

Original version

“... have distinct patterns in their azimuth angle (...) dependence (Fig. 1d and Fig. S1), consistent with the changes in the lattice structure being minor^{12,22}.”

Revised version

“... have distinct patterns in their azimuth angle (...) dependence (Fig. 1d, **Supplementary Fig. 1 and Table 1**), consistent with changes in the lattice structure being minor^{12,22}.”

#2. paragraph 8 in revised version

In response to reviewer #1 comment #2, we have added a counter-example as follows

Original version

“... . Thus, with all the lattice degrees of freedom exhausted, the QEP can only come from electronic scattering.”

Revised version

“... . **For example, the Raman spectra of LiOsO_3 show very minor change below 200 cm^{-1} across its second-order structural phase transition^{45,46}.** Thus, with all the lattice degrees of freedom exhausted, the QEP can only come from electronic scattering.”

#3. paragraph 10 in revised version

In response to reviewer #4 comment #4 and #5, we have revised as follows

Original version

"The phonon peaks are fitted with asymmetric Fano line-shape as shown in the inset. The resulting static susceptibility is not affected in any significant way by the fitting method because the phonon contributions are small."

Revised version

"The phonon peaks are fitted with asymmetric Fano line-shape as shown in the inset (see Supplementary Figs. 2 and 3 for the complete fitting result). The resulting static susceptibility is not affected in any significant way by the fitting method because the phonon contributions are small. The result is also in excellent agreement with that obtained using the standard coupled electron-phonon model (Supplementary Fig. 4)."

#4. paragraph 17 in revised version

In response to reviewer #4 comment #5, we have revised as follows

Original version

"Mode 5 is even more strongly damped and its weight is broadly distributed in the range $75 \sim 150 \text{ cm}^{-1}$ close to the T_c (Fig. 4). This mode has been predicted to be unstable in a recent DFT calculation²⁵ (see Supplementary Table 1 and Fig. S2)."

Revised version

"Mode 5 is even more strongly damped and its weight is broadly distributed in the range $75 \sim 150 \text{ cm}^{-1}$ close to the T_c (Fig. 4 and Supplementary Fig. 5). This mode has been predicted to be unstable in a recent DFT calculation²⁵ (see Supplementary Table 2)."

#5. related paragraphs in revised version

In response to reviewer #3 comment #3, we have added " B_{2g} " in front of "acoustic phonon" as follows

Original version

(paragraph 13)

"... the fact that excitonic fluctuations persist up to the highest measured temperature, in contrast to the softening of the acoustic phonon, ..."

(paragraph 14)

"(ii) The C_{55} elastic constant for the monoclinic strain (or the corresponding acoustic phonon velocity)"

(paragraph 17)

"... , the strong damping implies a rapid anharmonic decay into a pair of acoustic phonons of opposite momenta⁵³, The renormalization of the acoustic phonon dispersion to zero velocity at $\mathbf{q}=0$ (ref. 19) reduces the phase space for anharmonic phonon-phonon scattering, ..."

Revised version

(paragraph 13)

"... the fact that excitonic fluctuations persist up to the highest measured temperature, in contrast to the softening of the B_{2g} acoustic phonon, ..."

(paragraph 14)

"(ii) The C_{55} elastic constant for the monoclinic strain (or the corresponding B_{2g} acoustic phonon velocity)"

(paragraph 17)

"... , the strong damping implies a rapid anharmonic decay into a pair of B_{2g} acoustic phonons of opposite momenta⁵³, The renormalization of the B_{2g} acoustic phonon dispersion to zero velocity at $\mathbf{q}=0$ (ref. 19) reduces the phase space for anharmonic phonon-phonon scattering, ..."

#6. figure 1 in revised version

In response to reviewer #4 comment #2 and #3, we have revised as follows

Original version

“... navy (yellow) color represents the scattered-light polarization parallel (perpendicular) to the incident one.”

Revised version

“... navy (yellow) color represents the scattered-light polarization parallel (perpendicular) to the incident one. The A_q (B_{2q}) spectrum in the inset of c is isolated by measuring in the cross-polarization channel at $\psi=45^\circ$ ($\psi=0^\circ$).”

#7. figure 2, 3, and 4 in revised version

In response to reviewer #4 comment #3, we have indicated photon polarizations

Original version

“ B_{2g} ”

Revised version

“ B_{2g} (xz)”

#8. references in revised version

In response to reviewer #1 comment #2, we have added relevant reference as follows

- [45] Y. Shi, Y. Guo, X. Wang, A. J. Princep, D. Khalyavin, P. Manuel, Y. Michiue, A. Sato, K. Tsuda, S. Yu, M. Arai, Y. Shirako, M. Akaogi, N. Wang, K. Yamaura, and A. T. Boothroyd, *Nat. Mater.* **12**, 1024-1027 (2013)
- [46] F. Jin, L. Wang, A. Zhang, J. Ji, Y. Shi, X. Wang, R. Yu, J. Zhang, E. W. Plummer, and Q. Zhang, *Proc. Natl. Acad. Sci. U.S.A.* **116**, 20322-20327 (2019)

#9. section 1 in supplementary information

In response to reviewer #4 comment #2, we have added fitted Raman tensor elements

Supplementary table 1 (added)

Raman tensor elements. Fitting of the polarization dependence of the Raman modes, using the tensors given in Eq. (1) and (2). The results are overlaid with the data points in Fig. S1. The coefficients are normalized in such a way that $\|c_1\| + \|c_3\| + \|c_4\| = 1$.

#10. section 2 in supplementary information

In response to reviewer #4 comment #4 and #5, we have added a new section with relevant materials

Supplementary figure 2 (added)

Fitting result using the Fano line-shape (blue) and the damped harmonic oscillator line-shape (red). All measured spectra at temperatures between 280 K and 420 K are shown.

Supplementary figure 3 (added)

Fitting parameters for QEP. a, Electronic susceptibility in A_q (diamond) and B_{2q} (circle) channel. b, the inverse of B_{2q} electronic susceptibility. c, the inverse of QEP amplitude. d, the width of QEP. Error-bars indicate the standard deviation from the fitting procedure.

Supplementary figure 4 (added)

Fitting results for mode 2 with different fitting methods. a, QEP (red) fitted with the damped harmonic oscillator line-shape, and mode 2 (blue) with the Fano line-shape, respectively. b-e, amplitude, energy, width, and asymmetry of mode 2 from fitting method 1. f, QEP and mode 2 fitted with fitting method 2. Bare electronic feature (red) and mode 2 (blue) without electron-phonon coupling are shown. g-j, amplitude, energy, width, and asymmetry of mode 2 from fitting method 2.

#11. section 3 in supplementary information

In response to reviewer #4 comment #5, we have added a figure

Supplementary figure 5 (added)

Vertically stacked Raman spectra. a, B_{2g} Raman spectra measured in $-y(xz)y$. b, A_g in $-y(xx)y$. Black solid line indicates T_c , and the leakage of A_g mode 3 is marked by an asterisk. Mode 5 (inverted triangle) is heavily damped at T_c , but never softens to zero energy.

#12. references in supplementary information

In response to reviewer #4 comment #4 and #5, we have added relevant reference as follows

- [1] M. V. Klein, "Electronic Raman Scattering" in Light Scattering in Solids I, Vol. 8 (Springer-Verlag, Berlin, 1983) Chap. 4, pp. 147-202.
- [2] P. A. Volkov, M. Ye, H. Lohani, I. Feldman, A. Kanigel, K. Haule, and G. Blumberg, Critical charge fluctuations and quantum coherent state in excitonic insulator Ta_2NiSe_5 , arXiv:2007.07344 (2020).